# Updates on Epstein–Barr Virus (EBV)-Associated Nasopharyngeal Carcinoma: Emphasis on the Latent Gene Products of EBV

**DOI:** 10.3390/medicina59010002

**Published:** 2022-12-20

**Authors:** Naveed Ahmed, Mai Abdel Haleem A. Abusalah, Anam Farzand, Muhammad Absar, Nik Yusnoraini Yusof, Ali A. Rabaan, Hajir AlSaihati, Amer Alshengeti, Sara Alwarthan, Haifa S. Alsuwailem, Zainb A. Alrumaih, Ahmed Alsayyah, Chan Yean Yean

**Affiliations:** 1Department of Medical Microbiology and Parasitology, School of Medical Sciences, Universiti Sains Malaysia, Kubang Kerian 16150, Kelantan, Malaysia; 2Faculty of Medical Allied Science, Zarqa University, Zarqa 13110, Jordan; 3Department of Allied Health Science, Superior University, Lahore 54000, Pakistan; 4Institute for Research in Molecular Medicine (INFORMM), Health Campus, Universiti Sains Malaysia, Kubang Kerian 16150, Kelantan, Malaysia; 5Molecular Diagnostic Laboratory, Johns Hopkins Aramco Healthcare, Dhahran 31311, Saudi Arabia; 6College of Medicine, Alfaisal University, Riyadh 11533, Saudi Arabia; 7Department of Public Health and Nutrition, The University of Haripur, Haripur 22610, Pakistan; 8Department of Clinical Laboratory Sciences, College of Applied Medical Sciences, University of Hafr Al Batin, Hafr Al Batin 39831, Saudi Arabia; 9Department of Pediatrics, College of Medicine, Taibah University, Al-Madinah 41491, Saudi Arabia; 10Department of Infection Prevention and Control, Prince Mohammad Bin Abdulaziz Hospital, National Guard Health Affairs, Al-Madinah 41491, Saudi Arabia; 11Department of Internal Medicine, College of Medicine, Imam Abdulrahman Bin Faisal University, Dammam 34212, Saudi Arabia; 12Department of Medicine, College of Medicine, Princess Norah Bint Abdulrahman University, Riyadh 84428, Saudi Arabia; 13College of Medicine, Imam Abdulrahman Bin Faisal University, Dammam 34212, Saudi Arabia; 14Department of Pathology, College of Medicine, Imam Abdulrahman Bin Faisal University, Dammam 31441, Saudi Arabia

**Keywords:** Epstein–Barr virus, virus, carcinoma, nasopharyngeal carcinoma, oncogenic viruses

## Abstract

Nasopharyngeal carcinoma (NPC) is an uncommon type of malignancy/cancer worldwide. However, NPC is an endemic disease in southeast Asia and southern China and the reasons behind the underlying for such changes are unclear. Even though the Epstein–Barr infection (EBV) has been suggested as an important reason for undistinguishable NPC, the EBV itself is not adequate to source this type of cancer. The risk factors, for example, genetic susceptibility, and environmental factors might be associated with EBV to undertake a part in the NPC carcinogenesis. Normal healthy people have a memory B cell pool where the EBV persists, and any disturbance of this connection leads to virus-associated B cell malignancies. Less is known about the relationship between EBV and epithelial cell tumors, especially the EBV-associated nasopharyngeal carcinoma (EBVaNPC) and EBV-associated gastric carcinoma (EBVaGC). Currently, it is believed that premalignant genetic changes in epithelial cells contribute to the aberrant establishment of viral latency in these tumors. The early and late phases of NPC patients’ survival rates vary significantly. The presence of EBV in all tumor cells presents prospects for the development of innovative therapeutic and diagnostic techniques, despite the fact that the virus’s exact involvement in the carcinogenic process is presently not very well known. EBV research continues to shed light on the carcinogenic process, which is important for a more comprehensive knowledge of tumor etiology and the development of targeted cancer therapeutics. In order to screen for NPC, EBV-related biomarkers have been widely used in a few high-incidence locations because of their close associations with the risks of NPC. The current review highlights the scientific importance of EBV and its possible association with NPC.

## 1. Introduction

The Epstein–Barr virus (EBV), which is sometimes also referred to as human herpesvirus 4, is a member of the herpes virus family [1]. It is one of the most prevalent viruses in humans and may found worldwide. The majority of individuals become infected with EBV at some point in their life [2]. The EBV infection is frequently transmitted by bodily fluids, particularly saliva. EBV might be responsible for diseases such as infectious mononucleosis, and some other diseases [3,4].

Children are most often infected with EBV [5]. Most of the time, pediatric EBV infections do not cause symptoms, or if they do, those symptoms resemble those of other mild, temporary childhood disorders [2,6]. Usually in adults or teenage patients, those with EBV infection symptoms heal in two to four weeks. However, for some people, fatigue may last for many weeks or even months. After contracting EBV, the virus goes into a latent (inactive) state. The virus may sometimes reactivate. Although symptoms are not always present, those with compromised immune systems are more prone to have them if EBV reactivates [7]. Saliva in particular is a frequent bodily fluid through which EBV might be transmitted, but EBV may also be transmitted by organ transplants, blood transfusions, and through blood or semen during sexual intercourse. EBV can also be transmitted through items that an infected person recently used, such a toothbrush or drinking glass. The virus most likely persists on an item for at least as long as it is moist. The host might spread the virus for weeks or even before symptoms appear after contracting EBV for the first time (primary EBV infection) [8,9].

Both the eradication of viral infections and the inhibition of tumor development depend critically on the activation of inflammasomes and the subsequent inflammatory response [10]. Chronic inflammation, which promotes the growth of tumors and virus replication, is caused by abnormal production of proinflammatory cytokines as a result of dysregulated inflammasome signaling. Due to its inhibitory activity against ongoing inflammation, which may make diseases worse, and by giving a chance to increase the effectiveness of immune checkpoint blockade in cancer immunotherapy, targeted inhibition of inflammasome activity may have potential applications in cancer prevention and therapy [11,12]. Inflammasome-mediated antitumor immune responses are evaded by viruses using a variety of tactics, according to research published recently. The evasion mechanisms in EBV-associated malignancies, however, still need much more research [13]. Furthermore, nothing is known about the clinical function of NLR inflammasomes in response to EBV infection. Prior to this, the divergent impacts of inflammasomes in cancer further imply that there is more to learn about the functional roles and processes involved [14,15].

The genome of EBV is 172 kb long which encodes almost 100 open reading frames (ORFs) [16]. These ORFs are divided into lytic and latent genes. The latent proteins consist of three latent membrane proteins (LMP1, LMP2A, and LMP2B) and six nuclear antigens [17]. Through the regulation of many important cellular processes and pathways, these latent proteins have come to be recognized as essential components in the pathogenesis of EBV-associated malignancies [18]. Studies continue to be carried out extensively to determine how these latent proteins affect the oncogenic process. Although there are more than 80 EBV lytic genes, the potential involvement of the bulk of these genes in oncogenesis has received far less attention [19]. However, it is now becoming increasingly evident that the EBV lytic phase also contributes significantly to EBV-associated carcinogenesis. In particular, the constant expression of an early lytic gene, BARF1, in NPC and EBVaGC emphasizes the need to be more accepting of the potential role of EBV genes other than those often connected with latent infection [20]. The current review provides an overview on EBVaNPC, and the role of latent gene products of EBV in carcinogenesis.

## 2. Definition of Nasopharyngeal Carcinoma

A very rare type of carcinoma, which belongs to head and neck cancers, is nasopharyngeal carcinoma (NPC) [21]. It was also known as lymphoepithelioma previously. NPC originates in the nasopharynx which is an upper respiratory tract organ located behind the nose, above the back of throat and near to the Eustachian tubes [22]. Due to this complicated anatomic location and great radiosensitivity, for the treatment of non-metastatic or stage I NPC, radiation therapy (RT) is strongly advised [23]. However, the majority of patients only receive a diagnosis after the disease has progressed, since early-stage NPC symptoms such as headache, nosebleed, nasal blockage, and nasal discharge are benign. Southeast Asia has a higher prevalence of NPC cases, which are commonly but not always associated with EBV [24,25].

Based on how the tumor cells look under a light microscope, the World Health Organization (WHO) has classified NPC into two basic histological types. One is keratinizing squamous cell carcinoma (type I) and the other is non-keratinizing squamous cell carcinoma (type II and type III). Differentiated non-keratinizing carcinoma (type II) and undifferentiated carcinoma (type III), which are mostly EBV-positive tumors, are additional subtypes of the non-keratinizing type [26]. Less than 20% of NPC cases globally are well-differentiated keratinizing NPCs (type I), and this tumor type is rather uncommon or not present in the southern part of China. However, it has been shown that EBV is notably associated with the WHO type I, more differentiated form of NPC in the areas with a high frequency of undifferentiated NPC [27]. In NPC, the virus only lives in the tumor cells in a latent condition; it is not present in the lymphoid infiltrate around the tumor [28]. However, the interaction between the nearby carcinoma cells and the protruding lymphoid stroma seen in undifferentiated NPC seems to be essential for the continuing proliferation of malignant cells of NPC [29,30].

### 2.1. Staging of NPC

The American Joint Committee on Cancer (AJCC) re-evaluated the staging procedure in 2018 due to the variations in imaging technology and better results linked to optimal medication [25]. Recent recommendations have established the definitions for the tumor node metastasis (TNM) classification of malignant tumor staging as primary tumors (T), nodal metastasis (N), and distant metastasis (M) [24]. The staging of EBV has been elaborated in Figure 1.

### 2.2. Diagnostic Features of NPC

The early detection of NPC is challenging, possibly because it is difficult to inspect the nasopharynx and because the symptoms of NPCs might be mistaken for those of other, more prevalent illnesses [18,19]. A more precise immunofluorescence method was developed to recognize antigens against replicative antigens encoded by EBV, and this test confirmed the correlation between higher antibody titers against the membrane antigen (MA) and EBV-encoded viral capsid antigen (VCA) with NPC [31,32]. Later, using the DNA hybridization technique EBV DNA was identified in the samples from NPC tumors. The serological analysis revealed a correlation between the EBV antibodies titers and the NPC tumor stage, as well as the particular IgA levels of VCA as a potential visualization pointer. The in situ hybridizations identified EBV DNA in the NPC growth cells but not in cells such as lymphoid penetrating cells [33,34].

In view of the obvious explicitness of the connection between the degrees of IgA of VCA and NPC, a mass serological location program in the city of Wuzhou, China was conducted in 1980, where it was shown that EBV-explicit antibodies are valuable for NPC’s initial distinguishing proof [35]. The exploration showed the presence of IgA viral capsid antigen as early as 3 years and 5 months before the clinical conclusion of NPC. Southern exchange hybridization of terminal repeats (TRs) in the EBV genome in NPC showed that occupant viral genomes were monoclonal, proposing that EBV disease happens before the clonal development of the inhabitants in malignant growth cells [36,37].

Using electronic microscopy, the particles of herpes virus in a subset of tumor cells derived from Burkitt lymphoma (BL) were detected [6,38]. After a few years, the antibodies were found in the blood of African patients with BL who showed antigens delivered by BL cells. The serum of patients with post-nasal space carcinoma displayed comparable BL antigen antibodies [20]. These antibodies were distinguished in an enormous number of African and American patients. The significant prevalence of positive sera among patients with post-nasal area carcinoma highlights the necessity to differentiate comparable particles in cancer-prevalent areas [20,25].

### 2.3. Morphological Similarity of Undifferentiated Carcinomas of Nasopharyngeal Type (UCNT)

The discovery of EBV in UCNT has motivated many organizations to investigate undifferentiated NPCs [39]. Stomach UCNTs are consistently positive for EBV, but the other UCNTs have a lower association with EBV [40]. Previously, EBV has been identified in Chinese patients with thymic epithelial tumors but not in Western patients. Additionally, whereas Caucasian patients do not have an association between UCNTs of the salivary glands and EBV, Greenlanders and Chinese patients do. UCNTs from the uterus and breast are free from EBV in a few cases [41,42].

EBV is a two-fold strand DNA infection with a genome from 170 to 180 kB which contains almost 100 qualities for a dormant or lytic disease of host cells [21]. The viral genome remains episomal during the latest stage of the disease. It activates a number of latent genes (>10) that are used to control different cellular processes and make use of the host’s DNA polymerase enzymes for the process of DNA replication [43]. Meanwhile, during necessary cell death, lytic infection causes the production of >80 lytic proteins and the release of virus particles into the extracellular space [44]. Despite the fact that EBV is now categorized as a category I carcinogen, more than 90% of healthy persons are lifetime carriers of the illness. Only a few memory B cells in healthy people retain EBV latency after initial infection, which is regulated by the host’s immune system [45]. To start tumors and promote clonal growth of infected lymphoid and epithelial cells, the virus, however, causes specific epigenetic/genetic modifications or impairs the host immune system [46].

EBV is the main human oncogenic infection to be recognized. It is connected to classic Hodgkin’s lymphoma (HL), Burkitt lymphoma (BL), B cell lymphoma (BcL), NK/T cell nasal lymphoma, two sorts of epithelial carcinoma, gastric disease (e.g., GC), and cellular breakdown in the lung’s cells (e.g., NPC) [47]. For GC and NPC, 84,000 and 78,000 cases of the 200,000 newly reported cases of lethal cancers related to EBV are diagnosed every year, respectively [48]. Stomach malignant growths brought about by EBV are under 10% of all stomach diseases and are not endemic. In endemic locales such as Hong Kong and southern China, practically all non-keratinizing NPCs are consistently connected with EBV disease [49]. Studies in the last thirty years have laid out that EBV contamination and different hereditary irregularities are the motors of the threatening growth of NPC. It is accepted that NPC is a clonal dangerous cancer from a solitary inert contaminated progenitor cell caused by EBV [50]. Every EBV study supports this idea. Similar numbers of TRs are present in the episomal genomes of NPC cells, which rules out the possibility of progenitor cell inactive replication due to EBV. The unstoppable virions’ linearized genomes are circularized by irregular and discontinuous episome formation throughout the EBV lytic cycle and this results in contamination of epithelial cells, resulting in a varied number of TRs in each EBV episome within the infected cell [51,52].

## 3. Symptoms and Transmission

NPC may not cause any symptoms in its early stages [53]. Possible noticeable symptoms of NPCs include:➢Neck lymph node enlargement.➢Bleeding from the mouth or nose.➢Ear-related issues and hearing loss caused by the tumor’s closeness to the Eustachian tube, which causes obstruction and fluid collection in the middle ear.➢Sore throat.➢Difficulty breathing through the nose.➢Headache.

Oncogenic viral infections are thought to play a key role in the development of 20% of all types of human cancers, or about 2 million cases every year [54]. Infection’s function in cancer research reveals the fundamental apparatuses which can initiate neoplastic progression and points to healing and preventative options. The first human tumor virus to be discovered was Epstein–Barr virus (EBV), and it has revealed a lot about cancer etiology and it also has the usual antiquity of chronic herpes virus septicity [55]. EBVaNPC, in many cases, is the cause of illness and mortality in China. Despite NPC’s high public health impact in regions where NPC is identified and prevented, the NPC risk factors include eating highly salt-preserved seafood and tobacco use [56]. In China, EBV accumulates in undifferentiated NPC (most NPCs). Individuals develop EBV cancer even though it affects the majority of humanity, suggesting that EBV alone is not enough to cause cancer [57,58].

An IgA antibody against Epstein-1 (EBNA1/IgA) and VCA/IgA assessed by enzyme-linked immunosorbent assay (ELISA) has higher specificity, sensitivity, and positive predictive value than other techniques, according to a recent study [25]. According to EBV serological markers, individuals diagnosed as having a high risk of NPC can be offered fiberoptic endoscopy/biopsy to lower mortality [59]. New biomarkers, both cost-effective and labor-effective, are needed to identify high-risk groups to serve the NPC community better [60].

Regarding transmission:➢The most common method of EBV transmission is via bodily fluids, notably saliva. EBV may, however, be spread by blood and sperm during sexual activity, blood transfusions, and organ transplants.➢EBV may be transmitted by touching an infected individual’s recently used drinking glass or toothbrush. The virus is likely to live on a surface for as long as it stays moist.

When infected with EBV for the first time (primary EBV infection), the virus may spread for many weeks prior to the onset of symptoms [61]. The virus stays latent inside the body (inactive). One may possibly transmit EBV to others if the virus reactivates, regardless of how long it has been since the first infection [62]. NPC is an uncommon head and neck tumor that begins in the nasopharynx. The nasopharynx is positioned at the very back of the nose, near the Eustachian tubes. EBV is common but not always the cause of NPC in Southeast Asia [63].

## 4. Latent Gene Products of EBV

Both of the EBV latent gene products (the EBERs and LMP1) and homogeneous lengths of TRs were previously discovered in NPC and precancerous lesions [64]. It was suggested that the clonal latent EBV infection is a key contributor to the development of these virus-related cancers. Additionally, earlier genomic and functional studies demonstrated that a number of specific genetic alterations in the premalignant nasopharyngeal epithelium support a cellular switch to a state that maintains persistent latent EBV infection and predisposes people to NPC transformation (such as the inactivation of CDNK2A/p16 and tumor suppressors at chromosome 3p) [65,66]. In actuality, latent EBV infection persistence and latent viral gene production are required for NPC formation. Examples of type II latency programs discovered in NPC include LMP1, LMP2, EBNA1, EBER1/2, and BARF1, several splicing non-coding RNAs, and a variety of miRNAs expressed in BART regions. While latent genes such as LMP1 and LMP2 are expressed in various ways in the tumor or when the illness worsens, EBERs and EBNA1 are present in all cancer cells [67,68].

While the EBV genome is lost during the long-term passage of different NPC in vitro cell lines, the dormant EBV disease is ceaselessly distinguished in every growth cell in xenograft models from patients and the NPC clinical tissues in both the fundamental and principal repeating textures in essential and repeating cases [69,70,71]. The existence of an EBV episomal genome and the necessity for a very large number of viral characteristics for detrimental alteration are among the key characteristics of EBVaNPC. As indicated by overviews, numerous viral inert qualities have been found to contribute to the cancer genesis of NPC by providing an assortment of malignant growth attributes [72]. Throughout recent years, epithelial cell lines have been utilized to concentrate on the cancer-causing properties of these dormant quality items and their part in the carcinogenesis of NPC [73]. The replication and the mitotic isolation of EBV episomes, as well as the safeguarding of EBV genomes in inert contaminated cells, all rely upon EBNA1. Additionally, it is demonstrated that EBNA1 upholds cell endurance after DNA harm by instigating genomic insecurity and initiating a few cell qualities through transcriptional enactment [74,75].

EBNA1 is only one of the numerous non-polyadenylated RNAs found in certain EBV malignant growth cells, which incorporate EBER1 and EBER2. In latent infected epithelial cells, the EBERs self-replicate and bind to the ribosomal protein L22 to deliver ribonucleoprotein particles [76,77]. The complex then connects to the PKR, hindering apoptosis by FAS. Additionally, these non-coding RNAs have shown the ability to promote growth improvement by enhancing IGF-1 autocrine production and activating the NF-B pathway through RIG-1 and TLR3 flagging [78]. Numerous multi-locus, non-long coding records, and viral components from BAMH1 are produced by NPC cells, a member of the EBV family. EBV-encoded miRNAs, often referred to as miR-BARTs, target a variety of viral and cellular genes in order to promote cell proliferation, enhance invasiveness, inhibit apoptosis, create genomic instability, and disrupt the apoptotic host immune response [79].

### 4.1. Expression of EBV Latent Gene in Virus-Associated Tumors

The interaction of EBV with B cells is essential to understand the biology of the virus. EBV can easily infect and modify B cells to a normal state in vitro tests, making them a useful model system for understanding the biology and behavior of the virus [80,81]. The LCL produced from in vitro B cells contains a modest number of latent proteins, including six nuclear antigens (EBNA 1, 2, 3A, 3B, 3C, and LP) and three latent membrane proteins (LMPS 1, 2A, and 2B). The LCL expresses the small RNAs coded by EBV EBER1 and EBER2, which are found in all types of infection with latent EBV and have proven useful to identify EBV in malignant tumors [82,83]. The LCL also includes BAMH1-A transcriptions to the right (BARTs) and transcripts from the reading region to the right BAMH1 (BHRF1) of the viral genome, which code for the microarc (Miarn). The form of latency III of EBV infection is identified by this model of expression of the latent EBV gene. Most of the lymphomas associated with EBV that occur in immunocompromised people include it [84,85].

EBV latent gene expression investigations in malignant tumors associated with the virus and cell lines created from BL biopsies have uncovered two extra EBV inertness qualities [17,86]. The EBER and BAMH1-A records include EBNA1, and the confined EBV protein is dependably recognized in BL EBV-positive cancers. This sort of dormancy is known as latency I. In around 5% to 10% of BL EBV-positive tumors, a kind of latency I with BHRF1 and EBNA 3A, 3B, and 3C is seen. The selection pressure to down-regulate the expression of EBNA2 in these tumors has occurred through deletion of the EBNA2 gene, as opposed to the switch in viral promoter usage observed in the conventional BL situation [87]. Latency II, a different kind of EBV latency, was first discovered in NPC biopsies and later discovered in instances of EBV-related HL. Here, LMP1 and LMP2A/B are also expressed together with the transcripts for EBERs, EBNA1, and BamHI-A. Only around 20% of biopsies had LMP1 protein levels that were unequivocally positive in NPC, while this latency II pattern of EBV latent gene expression is a recurrent feature of virus-associated HL [88,89,90].

The preferred method to identify EBV infection in cells and tissues is in situ hybridization to the many EBV-encoded short RNA (EBER) transcripts (top panel, left) [91]. NPC’s immunohistochemical staining indicates that every tumor cell expresses EBNA1 (right, upper panel). Latent membrane protein 1 (LMP1) and LMP2A expression levels in NPC biopsies (bottom panels) are more varied. It is thought that the significant lymphoid infiltration in NPC helps the tumor cells proliferate and survive [92].

The many types of EBV latent infections reflect the cellular milieu as well as the intricate interactions and cooperation between host regulatory elements and viral promoters that result in EBV latent features [93]. Differential grafting of the equivalent long (in excess of 100 kb) element translated to one side communicated from one of the two promoters (Wp or Cp) and found the other in the BAMH1-C and BAMH1-W locales of the genome in latency III, coding a few EBNAs, producing individual mRNAs. Towards the beginning of the B cell disease, the transactive activities of EBNA1 and EBNA2 on CP trigger a difference in Wp to Cp [94]. LMP1 mRNAs are produced from a distinct promoter in the BAMH1-n region of the EBV genome, while LMP2B is produced from a promoter that is similar to that of the bidirectional promoter (ED-L1) and reacts to the transactivation of EBNA2 as well [95,96].

The LMP2A promotor is additionally controlled by EBNA2. The transcripts require the circularization of the genome on the grounds that LMP2A and LMP2B cross the TRs in the U1 region [18]. The circularization is created by homologous recombination of the TRs, causing combined terminals that are extremely lengthy. Since the TRs converge with an indistinguishable number of reiterations including expansions of a remarkable contaminated progenitor cell, this property was utilized to evaluate the EBV clonality. However, the EBV clonality after the disease could be because of the specific development presented by the ideal articulation of LMP2A on a predetermined number of TRs [87].

In the relatively limited EBV latency found in NPC and HL, the promoter QP Sans tata promotes transcription of EBNA1. The TRs that govern the expression of LMP1 include another promoter called L1-TT-TR [47]. The numerous transcription programs that EBV uses when it travels the response of B cells from the germinal center and finally colonizes the B cell compartment with rest memory are reflected in the different forms of latency of EBV [97]. Pressures from the environment of the host and selection of cells associated with neoplastic growth are supposed to play a role in establishing erroneous viral transcription programs in malignant tumors linked to EBV [98,99].

### 4.2. Targeting the EBV Latent Proteins

Virally coded proteins LMP1, LMP2, and EBNA1 could be taken advantage of as restorative focuses in NPC. The capacity of EBNA1 has been studied because of its ceaseless articulation in every growth cell and the basic function in keeping up with the EBV episomal genome [13]. In light of its consistent articulation and its organic significance in the conservation, replication, and isolation of viral DNA all through viral dormancy and lytic reactivation, the EBNA1 protein is a vital object. As indicated by the examinations conducted throughout the last 10 years, EBNA1 is a medication protein, and particular medications focusing on the DNA contact are or the dimerization interface have shown viability in vivo. Besides the likenesses between the EBNA1 epitopes and the normal human antigenic focuses of the lupus, the protein arrangement EBNA1 N has no connection to the cell protein of the host [100,101].

### 4.3. Therapeutic Targeting of EBNA1

It is currently realized that EBNA1 connects with explicit host cell parts to cause viral inactivity and prompt host cell oncogenic change. With an ongoing leap forward in how we might interpret EBNA1′s primary science, new proof proposes that both EBNA1 dimers and oligomers assume a part in viral latency instructions [102]. In previous studies, the manners in which EBNA1 capacities may be disrupted and the feasibility of focusing on EBNA1 for the treatment of EBV-associated malignancies were explored [68,103]. The strategy for EBV targeted therapy is shown in Figure 2.

### 4.4. Latent EBV Genes’ Expression in NPC and Their Function

Analysts should initially comprehend the capacity of EBV latent qualities communicated in NPC to assess the role of viral disease in the malignant growth process [90]. Likewise, complete information on the capacity of these EBV inactive qualities could prompt the formation of creative symptomatic and helpful strategies [72].

#### 4.4.1. LMPs and EBNA1

The EBNA1 protein is essential for the separation and replication of the EBV epi-some, but it has also been shown to protect apoptotic cells, promote cell endurance, and directly contribute to the tumorigenic aggregate [104]. The destabilization of P53, the interference of the atomic assemblages of promyelocytic leukemia (PML), and the control of a few signaling pathways are among the outcomes [19,82].

#### 4.4.2. LMP1

In rodent fibroblast transformation tests, LMP1 is a regular oncogene, and it is important for EBV-prompted B cell change in vitro. When LMP1 is generated in cells, it up-regulates antiapoptotic proteins (Bcl-2, A20) and increases cytokine production (interleukin (IL)-6, IL-8), among other things [20]. LMP2A and LMP2B, the two LMP2 quality proteins, share a short cytoplasmic C-end and 12 hydrophobic film crossing gaps. Regardless of this, it appears that the cytoplasmic N-terminal district of LMP2A’s exceptional invulnerable receptor depends on the initiation of tyrosine and is necessary for the protein’s practical function. LMP2A appears to support B cell expansion and endurance without B cell receptors (BCRs). EBV’s penchant to attack memory B cells could make sense of this effect. Compared to LMP1, LMP2A articulation in NPC is steadier. LMP2A mRNA articulation was seen in excess of 98% of NPC cases when utilizing reverse transcriptase polymerase chain reaction (RT-PCR), however, LMP2B articulation was lower and resembled that of LMP1. LMP2A protein articulation has been described in over half of NPC cases, as per immunohistochemical methods [18].

#### 4.4.3. BamHI-A and EBER Regions

Two non-coding or non-polyadenylated RNAs, known as EBERs 1 and 2, are highly expressed in all kinds of EBV latency and act as sensitive indicators of EBV infection in cells and tissues [105]. The protein kinase RNA-activated (PKR), stable ribonucleoprotein fragments including autoantigen La and ribosomal protein L22, and inducible interferon are all targets of EBERs. Other viral genes, including as LMP1 and LMP2A/B, may block the generation of type-I interferon, which is caused by the interaction of EBERs with retinoic acid inducible gene-1. In NPC cell lines, EBERs also promote a growth factor resembling insulin-like growth factor-1. This impact has been related to improved proliferation in NPC cell lines, and the presence of IGF-1 in tumor biopsies supports its importance in NPC [72,92].

#### 4.4.4. EBV Strain Variation

When the genomes of EBV isolated from diverse parts of the globe or individuals with various virus-associated diseases are examined, restriction fragment length polymorphism analysis reveals striking similarities [106]. Different EBV isolates, nevertheless, show differences in the repetitive sections of the EBV genome. Gross deletions were discovered when the EBV genome from multiple BL cell lines was analyzed; some of these deletions explain biological differences. In particular, the gene encoding EBNA2 has been deleted in the non-transforming P3HR-1 virus [107]. The EBV genome’s EBNA2-encoding (BamHI-WYH) region allows for strain diversity, allowing all viral isolates to be divided into type 1 (EBV-1, B95.8-like) and type 2 categories (EBV-2, Jijoye-like). Due to this chromosomal variance, the EBNA2 protein may exist in two antigenically different versions that only share 50% amino acid similarity [74].

A subset of latent genes, particularly those encoding EBNA3A, EBNA3B, EBNA3C, and EBNA-LP, display allelic polymorphisms that are connected to the EBV type (with a succession homology of 50–80% depending upon the area) [72]. EBV-2 protection has less effective test results in vitro than disengaged EBV-1, which has practical results. Type 1 infection disconnects are prevalent (however not solely) in a few Western countries, as per viral confinement and seroepidemiological research [70]. The two variations, however, are bountiful in New Guinea, tropical Africa, and maybe other areas. Within each type, there is a change, as well as a significant distinction between EBV-1 and -2 [7]. The singular strains were distinguished by contrasting them and B95.8, which included anything from special base changes to gigantic deletions. While it was commonly imagined that infection with a few EBV strains must be found in immunocompromised people, another review demonstrates the way that seropositive individuals can be infected with numerous EBV types, segregated by adjustment of amount and presence after some time [63].

The coinfection of the host with many virus strains can help the evolution of EBV by allowing genetic recombination and variation [8]. In HIV-positive individuals and the Chinese population, this interspecific recombination has been seen. During the EBV multiplication induced by immunosuppression, it seems to be produced by the recombination of multiple EBV strains [12].

## 5. Primary Gene Product (Genetic Variability)

EBV infects B lymphocytes at the start of primary infection, but it is not clear where these cells are infected or if the infection also affects the higher respiratory epithelial cells. EBV colonizes the pool of memory B cells to develop long-term persistence in vivo. A latent infection is created, which is identified by the expression of a small subset of viral genes known as latent genes [43]. Different types of EBV latency exist. The virus mimics latency models seen in certain malignant tumors linked to EBV and utilizes them for various viral life cycle stages [11]. EBV must also multiply periodically during its life cycle to generate infectious virus for transmission to a new sensitive host, with epithelial regions in the oropharynx and salivary glands showing the main viral replication sites [108]. Thus, the natural history of chronic EBV infection is comparable to that of another herpesviruses. To express the latent and replicative forms of the viral life cycle, many cell lines are required [109].

When looking at tumor types connected to the virus, the dual tropism of EBV infection is clear. The in vivo memory B cell pool may be infected by EBV. In vitro, inactive B cells can be quickly transformed into infected lymphoblastoid cell lines (LCLs), which resemble several malignant B cell tumors closely associated with EBV [101]. However, the undifferentiated type of nasopharyngeal cancer has a strong association with EBV (NPC). EBV is also present in a subset of gastric adenocarcinomas and other carcinomas of the salivary glands. As a result, EBV epithelial infection can cause malignant transformation, requiring specific preventive and therapeutic treatments [74,76].

### EBV-Related Head and Neck Cancers (Adoptive VST Therapy)

The adoption of cytotoxic T cells specific to EBV as a complement to standard treatment to prevent and effectively treat NPC cells positive for EBV expressing subdominant EBV antigens (EBNA1, LMP1/2) has been proposed [37] and, therefore, they could be identified, activated, and multiplied for immunotherapeutic purposes. Many recent studies have shown that the use of cytotoxic T cells specific to EBV can help prevent and treat malignant and coupled tumors linked to EBV such as NPC [22,51].

The principal cytotoxic T cell was well defined for EBV and was used to develop the cytotoxic cells for patients with HLs. As per the outcomes, the mixture of these cells in patients prompted clinical antiviral action in vivo and brought down EBV DNA levels in their blood [38]. Recently, augmented autologous T cells from patients with lymphomas related to EBV, which were well defined for LMP1 and LMP2 qualities, were studied. They showed that these T cells, well defined for expanded EBV, were fit for actuating well-established total reactions in these people with not many aftereffects [49].

A stage I clinical preliminary trial uncovered that the treatment of recurrent NPC patients with ATB-explicit TB cells prompted the reduction of tumoral clinical reactions in 6 out of 10 patients. Meanwhile, scientists have observed that the receptive exchange of CTL well defined for autolysis was protected and connected to huge anticancer action in patients with cutting-edge NPC in a review containing ten patients [6]. In an investigation of 24 NPC patients positive for EBV, EBV-explicit T cells were effectively developed from 16 patients with metastatic types of EBV-positive NPC (72.7%) [26]. The utilization of these EBV-explicit T cells has brought about long-term clinical enhancements without significant adverse consequences. One more clinical preliminary trial of stage I/II concentrated on the impact of EBV-explicit T cells in patients with unmanageable NPC and found that they introduced a reduction of growth action [8]. Notwithstanding, there was just a base clinical reaction on account of metastatic NPCs. A total of 35 patients with recurrent or metastatic NPC progression were first treated with chemotherapy, then a receptive transfer of EBV-explicit T cells in a new stage II clinical preliminary trial. This brought about a reaction rate of 71% and an increment of 63% in endurance rates [12].

The utilization of receptive cell treatment has been carried out previously which designated the LMP1/2 and EBNA1 antigens that are communicated in NPC. The process made explicit LMP/EBNA1 T cells utilizing the ADE1-LMP poly adenovirus vector, which worked in the ideal improvement of viral T cells from low-recurrence antecedents [110]. Most patients with EBV-positive NPCs can create explicit LMP/EBNA1 cells. The adjustment of the NPC was connected to the number of explicit T cells of LMP/EBNA1 directed to the patient, as per their outcomes. This panel likewise showed that the allogenic creation out of the space of explicit T cells of LMP/EBNA1 expanded their recurrence and their power so they can be utilized clinically to treat NPCs [111]. This conclusion proposes that the receptive exchange of specific T cells of EBV has a promising clinical outcome in EBV-positive NPC patients and ought to be considered as a corresponding treatment to conventional NPC medicines, particularly in patients with a common or metastatic sickness who are less sensitive to chemotherapy [112,113].

## 6. Risk Factors

The etiology of NPCs has been a mystery for more than a century since it was initially reported in 1901. Studies of southern Chinese migrants demonstrate that the incidence of NPCs is 10 to 30 times higher than that of other races when they settle in other countries, an unusual trend among malignant tumors that suggests a heredity component of the risk of NPC [111]. Contrasted with local Israelis and local Swedes, North African outsiders in Israel and Sweden had a higher recurrence of NPC. Despite the fact that around half of individuals living in China or moving to Southeast Asia have NPCs, the recurrence is significantly higher among Chinese born in Western nations than Caucasians [77]. Likewise, contrasted with French men born in the south of France, French men born in North Africa had a higher risk of NPC [108].

A family history of NPCs and some natural variables appear to assume a vital role in NPCs, as shown by the most recent examinations. The risk factors include Cantonian ethnicity, male sex, EBV disease, routinely eating fish preserved in salt, low utilization of new vegetables and natural products, smoking, and certain antigen classes of human leukocytes (HLA) [111]. Other HLA genotypes, such as with a history of infectious mononucleosis (IM), can be related to a decrease in risk. History of chronic respiratory illnesses, high intake of other preserved foods, and genetic polymorphisms in cytochrome CYP2A6, P450 2E1 (CYP2E1), GSTT1, and glutathione S-transferase M1 (GSTM1) are additional possible risk factors. Herbal drug use, occupational exposure to dust and formaldehyde, and nickel exposure are less well-established risk factors [77].

## 7. Conclusions

The prevalence of EBV infection demonstrates the close relationship between the virus and immune system, which is characterized by a persistent, asymptomatic infection that resides in the memory B cell compartment. EBV-associated B cell malignancies originate from perturbing this connection, which happens in different types of immunosuppression. Although it seems to be a consequence of the virus’s improper establishment of latency in epithelial cells that have previously undergone premalignant genetic modifications, the virus’s function in the link between EBVaNPC and EBVaGC is less clear. Regardless of the intricacies of these interactions and the precise role that EBV plays in the carcinogenic process, there is certainly an opportunity to exploit this relationship for the therapeutic benefit of patients. We may successfully target EBV-associated carcinomas if we use novel treatment strategies including viral reactivation, gene therapy, or therapeutic vaccination.

## Figures and Tables

**Figure 1 medicina-59-00002-f001:**
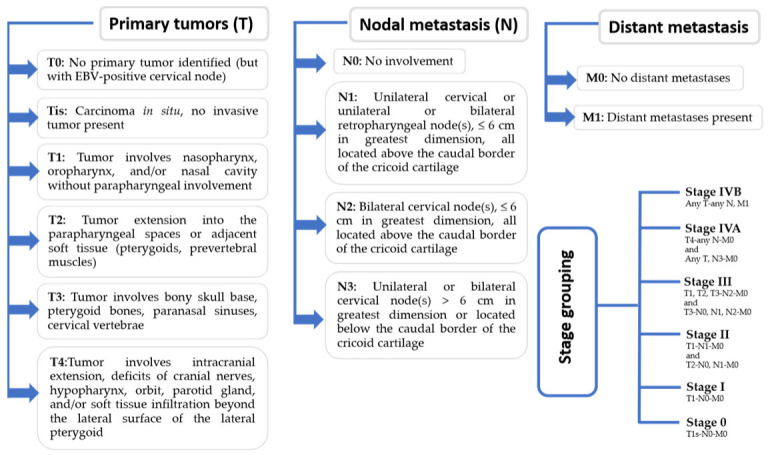
The tumor node metastasis (TNM) staging for nasopharyngeal carcinoma (NPC).

**Figure 2 medicina-59-00002-f002:**
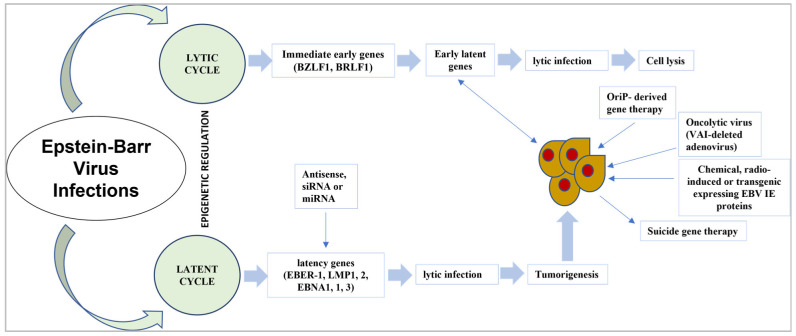
Strategy for EBV targeted gene therapy.

## Data Availability

Not applicable.

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
