# Peer review of "Updates on Epstein–Barr Virus (EBV)-Associated Nasopharyngeal Carcinoma: Emphasis on the Latent Gene Products of EBV"

_medicina, 2022, doi:10.3390/medicina59010002_

Round 1
Reviewer 1 Report
“Clinical Importance of Epstein–Barr Virus-Associated Nasopharyngeal Carcinoma: A Review”, highlights the scientific importance of EBV and its possible association with NPC. Although many studies indicate EBV infection is associated with NPC malignancy, but so far there is still no clear evidence in clinic.
Thus, Naveed Ahmed et al. organize and update the clinical studies of NPC with EBV, and discuss how EBV infection facilitates increase of NPC malignancy.
However, they discuss the latent genes of EBV may influence NPC malignancy, but lytic genes of EBV may also affect NPC malignancy. Naveed Ahmed et al. must further discuss how lytic genes of EBV affect NPC malignancy.
Because Naveed Ahmed et al. provides that the latent genes of EBV are involved in regulating NPC malignancy. Naveed Ahmed et al. can also support whether the drugs which can inhibit the EBV infection (or latent genes) applied in clinic.
Finally, Naveed Ahmed et al. may change their topic to emphasize their key points in this review article.
Author Response
Reviewer 1
Comments and Suggestions for Authors
“Clinical Importance of Epstein–Barr Virus-Associated Nasopharyngeal Carcinoma: A Review”, highlights the scientific importance of EBV and its possible association with NPC. Although many studies indicate EBV infection is associated with NPC malignancy, but so far there is still no clear evidence in clinic. Thus, Naveed Ahmed et al. organize and update the clinical studies of NPC with EBV, and discuss how EBV infection facilitates increase of NPC malignancy. However, they discuss the latent genes of EBV may influence NPC malignancy, but lytic genes of EBV may also affect NPC malignancy. Naveed Ahmed et al. must further discuss how lytic genes of EBV affect NPC malignancy. Because Naveed Ahmed et al. provides that the latent genes of EBV are involved in regulating NPC malignancy. Naveed Ahmed et al. can also support whether the drugs which can inhibit the EBV infection (or latent genes) applied in clinic.
Response: Dear reviewer, we would like to appreciate your efforts for reviewing the current manuscript. We really appreciate that you have liked our work and recommended for possible publication in “Medicina”. Furthermore, thank you for your valuable suggestion to add the literature about lytic gene. However, our main objective was to give an overview for EBVaNPC and the latent gene products. To make it clarify, we have revised the title in order to be strict on latent genes only. We understand the importance of Lytic gene, and we are already writing a separate detailed review article on this to cover the lytic gene aspects of EBVaNPC. Hence, we would like to request you to allow us to proceed with the latent gene details only to stay strict on the aims of review.
Finally, Naveed Ahmed et al. may change their topic to emphasize their key points in this review article.
Response: Dear reviewer, thank you for your valuable suggestion. The title of the manuscript has been amended.
Reviewer 2 Report
The authors are focused on the Ebstein Barr virus and effects on the nasopharyngeal carcinoma. They showed enormous scientific effort and diligent work to elucidate the effect of the virus on nasopharyngeal cancer. The title wakes up the attention immediately and the abstract even strengthen the interest. The figures and graphs are sufficient to draw the image of the research. The literature covers all known fields of knowledge. I would recommend accepting the article for publication.
Author Response
Reviewer 2
Comments and Suggestions for Authors
The authors are focused on the Epstein Barr virus and effects on the nasopharyngeal carcinoma. They showed enormous scientific effort and diligent work to elucidate the effect of the virus on nasopharyngeal cancer. The title wakes up the attention immediately and the abstract even strengthen the interest. The figures and graphs are sufficient to draw the image of the research. The literature covers all known fields of knowledge. I would recommend accepting the article for publication.
Response: Dear reviewer, we would like to appreciate your efforts for reviewing the current manuscript. We really appreciate that you have liked our work and recommended for possible publication in Medicina.